# Tamoxifen-resistant breast cancer cells exhibit reactivity with *Wisteria floribunda* agglutinin

May Thinzar Hlaing[1], Yoshiya Horimoto[1,2]*, Kaori Denda-Nagai[3], Haruhiko Fujihira[3,4], Miki Noji[3], Hiroyuki Kaji[5], Azusa Tomioka[5], Yumiko Ishizuka[1], Harumi Saeki[2], Atsushi Arakawa[2], Mitsue Saito[1], Tatsuro Irimura[3]

1 Department of Breast Oncology, Juntendo University Faculty of Medicine, Tokyo, Japan, 2 Department of Human Pathology, Juntendo University Faculty of Medicine, Tokyo, Japan, 3 Division of Glycobiologics, Department of Breast Oncology, Juntendo University Faculty of Medicine, Tokyo, Japan, 4 Glycometabolic Biochemistry Laboratory, Cluster of Pioneering Research, RIKEN, Saitama, Japan, 5 Molecular & Cellular Glycoproteomics Research Group, Cellular & Molecular Biotechnology Research Institute, National Institute of Advance Industrial Science & Technology (AIST), Tsukuba, Ibaraki, Japan

* horimoto@juntendo.ac.jp

**Data Availability Statement:** All relevant data are within the manuscript and its Supporting Information files.

## Abstract

Glycosylation is one of the most important post-translational modifications of cell surface proteins involved in the proliferation, metastasis and treatment resistance of cancer cells. However, little is known about the role of glycosylation as the mechanism of breast cancer cell resistance to endocrine therapy. Herein, we aimed to identify the glycan profiles of tamoxifen-resistant human breast cancer cells, and their potential as predictive biomarkers for endocrine therapy. We established tamoxifen-resistant cells from estrogen receptor-positive human breast cancer cell lines, and their membrane-associated proteins were subjected to lectin microarray analysis. To confirm differential lectin binding to cellular glycoproteins, we performed lectin blotting analyses after electrophoretic separation of the glycoproteins. Mass spectrometry of the tryptic peptides of the lectin-bound glycoproteins was further conducted to identify glycoproteins binding to the above lectins. Finally, expression of the glycans that were recognized by a lectin was investigated using clinical samples from patients who received tamoxifen treatment after curative surgery. Lectin microarray analysis revealed that the membrane fractions of tamoxifen-resistant breast cancer cells showed increased binding to *Wisteria floribunda* agglutinin (WFA) compared to tamoxifen-sensitive cells. Glycoproteins seemed to be responsible for the differential WFA binding and the results of mass spectrometry revealed several membrane glycoproteins, such as CD166 and integrin beta-1, as candidates contributing to increased WFA binding. In clinical samples, strong WFA staining was more frequently observed in patients who had developed distant metastasis during tamoxifen treatment compared with non-relapsed patients. Therefore, glycans recognized by WFA are potentially useful as predictive markers to identify the tamoxifen-resistant and relapse-prone subset of estrogen receptor-positive breast cancer patients.

**Funding:** The authors received no specific funding for this work.

**Competing interests:** The authors have declared that no competing interests exist.

## Introduction

Breast cancer is one of the most prevalent diseases in women in Japan, where the incidence and mortality rate has been increasing annually [1]. Based on hormone-receptor (HR) and human epidermal growth factor receptor-2 (HER2) status, breast cancer is classified as either HR-positive HER2-negative, HR-positive HER2-positive, HER2 type, or triple-negative breast cancer [2]. For patients with HR-positive breast cancer, an adjuvant endocrine therapy (ET) will be administered for five to ten years following curative surgery [3, 4]. ET includes selective estrogen receptor modulators, selective estrogen receptor downregulators and aromatase inhibitors, and treatment is based on the patient's age and side effects. Although early detection techniques, therapeutic regimens and understanding of the molecular basis of breast cancer biology has been advancing, nearly 30% of all patients with early stage breast cancer have recurrence, and most cases are metastatic [5]. *De novo* and acquired resistance to these therapies is one of the major causes of breast cancer mortality [6] where several molecules may contribute to the acquisition of ET resistance, such as those in the PI3K/mTOR and RAS/RAF/MEK/ERK pathways [7, 8]. Epigenetic alterations can also facilitate the escape of tumor cells from ETs [9]. The combination of hormone therapy with CDK4/6 inhibitors and selective inhibitors of PI3K and mTOR can increase progression-free survival in clinical practice [10]. Although treatments to overcome recurrence and resistance of breast cancer are constantly being developed, the mechanism of resistance is still difficult to understand. A new approach that does not employ the conventional gene and protein expression analysis might be required to uncover these points of resistance.

Glycosylation is a major post-translational modification of proteins through the sequential actions of glycosyltransferases [11, 12]. Alteration of glycan is known to be associated with carcinogenesis, malignant progression and metastasis [13]. Glycosylation changes have also been implicated in drug resistance of cancer. Multidrug resistance in cancer is mainly caused by high expression of P-glycoprotein [14]. For example, profiling of a cisplatin-resistant ovarian cancer cell line with a lectin array and a gene expression array revealed higher expression of core fucose, poly *N*-acetyllactosamine, and high mannose structures compared with the parental cells [15]. Furthermore, resistance to paclitaxel and other chemotherapeutic agents are associated with altered glycans in breast cancer [16, 17]. However, there have been no studies focused on the role of glycosylation in endocrine resistance in HR-positive breast cancer cells. Here, we report the presence of glycans recognized by *Wisteria floribunda* agglutinin (WFA) in tamoxifen (TAM) resistant HR-positive breast cancer cells and surgical specimens from patients who exhibited early recurrence after TAM treatment.

## Materials and methods

### Establishment of TAM-resistant cells

The human breast cancer cell lines T47D and ZR75-1 were purchased from the American Type Culture Collection. They were cultured in Gibco Dulbecco's Modified Eagle's Medium (Thermo Fisher Scientific) with 10% fetal bovine serum and 100 U/mL penicillin/streptomycin in a humidified incubator with 5% $CO_2$ at 37°C.

TAM (4-hydroxytamoxifen, H7904) was purchased from Sigma-Aldrich and TAM-resistant cells (T47D-TAM$^R$ and ZR75-1-TAM$^R$) were established by chronic exposure of the drug-sensitive T47D and ZR75-1 cells to stepwise increases in TAM concentrations, until a resistance concentration was achieved. The resistance to TAM in T47D-TAM$^R$ and ZR75-1-TAM$^R$ cells was confirmed by a Cell Counting Kit-8 (Dojindo Laboratories) which showed a higher viability of both cell types after TAM treatment compared with the sensitive cells (S1

Fig). The LD50 (50% lethal dose) in T47D and T47D-TAM$^R$ cells was 4.0 μM and 6.3 μM, respectively, and in ZR75-1 and ZR75-1-TAM$^R$ cells was 9.5 μM and 10.4 μM, respectively.

## Membrane protein extraction

Pellets containing $1x10^7$ cells were collected and the membrane protein (hydrophobic fraction) was extracted using CelLytic$^{TM}$ MEM Protein Extraction Kit (Sigma Aldrich) according to the manufacturer's protocol. The protein concentration of the obtained hydrophobic fraction was quantified with a Pierce$^{TM}$ BCA assay kit (Thermo Scientific).

## Lectin microarray analysis

Membrane proteins were diluted with phosphate buffered saline (50 μg/mL) and 20 μL aliquots were labelled with 100 μg of Cy3 succinimidyl ester (GE Healthcare) and incubated at room temperature for 1 h in the dark. Excess labeling reagent was removed by spin column (Zeba spin, Thermo Fisher Scientific), and the treated Cy3-labeled samples diluted with probing solution to 500 ng/mL (GlycoTechnica Ltd). Aliquots (100 μL) of the prepared Cy3-labelled samples were applied to the LecChip (GlycoTechnica Ltd) and incubated overnight at 20˚C with gentle shaking, to allow for complete formation of the lectin-glycan complex. Then, the lectin microarray was washed three times with the Probing Solution and scanned with a GlycoStation$^{TM}$ Reader 1200 (GlycoTechnica Ltd). Image scanning was performed at the highest net intensity for 45 lectins, around 50,000 arbitrary fluorescence units. Fluorescence intensities were analyzed using GlycoStation$^{TM}$ Signal Capture Ver. 1.5 and GlycoStation$^{TM}$ ToolsPro Suite Ver. 1.5 (GlycoTechnica Ltd). The average intensity of the three spots for each lectin were normalized using the average value of 45 lectins. The normalized average value was compared between the sensitive and resistant breast cancer cell lines.

## Lectin blotting

Membrane protein fractions (10 μg) were loaded per lane, separated by 10% sodium dodecyl sulfate–polyacrylamide gel electrophoresis (SDS-PAGE), and blotted on a polyvinylidene difluoride membrane (Millipore). The membrane was blocked with Bullet Blocking One (Nacalai Tesque) and incubated with biotinylated WFA (2.5 μg/mL; for preparation of biotinylated WFA, please refer to "Clinical samples and lectin staining" below) and biotinylated *Sambucus nigra* agglutinin (SNA) (2.5 μg/mL) (Vector Laboratories) at 4˚C overnight. Membranes were subsequently incubated with horseradish peroxidase conjugated streptavidin (1:4000 dilution, Jackson ImmunoResearch) for 1 h at room temperature. The bands were visualized using Clarity Western ECL detection reagent (Bio-Rad Laboratories) and detected by Chemi-Doc Touch (Bio-Rad Laboratories).

## Preparation of WFA-binding proteins and identification by liquid chromatography/mass spectrometry (LC/MS) followed by database search using Mascot

For lectin precipitation, prepared hydrophobic fractions (50 μg) diluted in Milli-Q water (up to 40 μL) were subjected to immunoprecipitation using biotinylated WFA. The membrane proteins were preheated at 95˚C for 5 min and precleared with 20 μL of Dynabeads My One Streptavidin T1 (Dyna-SA) (Veritas) for 2 h at 4˚C. Biotinylated WFA was reacted with Dyna-SA in 20 μL of Tris-buffered saline (TBS) containing 1% Triton-X (TBSTx) for 2 h at 4˚C. The beads conjugated with WFA were washed with TBSTx, and precleared membrane proteins were added to the beads that continued overnight incubation at 4˚C. The supernatant was

removed, and the beads were washed with TBSTx. Washed beads were suspended into 40 μL of TBS containing 0.2% SDS and incubated at 95°C for 10 min to elute the precipitated WFA-binding proteins. Supernatants were collected and shaken with 20 μL of Dyna-SA at 4°C for at least 60 min for depletion of biotinylated WFA which was eluted in the elution step. The samples were centrifuged at 13,000 rpm for 1 min at 4°C and the supernatants were collected and used as the WFA binding fraction.

SDS-PAGE was performed using 10% polyacrylamide gel under a constant current of 30 mA for 53 min. The gel was silver stained (AE-1360 EzStain Silver, Atto), and a band around 100 kDa, corresponding to the bands detected by lectin blotting using WFA, was excised. The cut gel was decolorized for 15 min at room temperature using reagents from the Silver Stain MS kit (Fujifilm Wako). After washing the gel five times with MilliQ water (500 μL each time), the gel was twice washed with acetonitrile (200 μL each time) to dehydrate, and then dried. In-gel digestion was carried out as described previously [18].

Peptide mixture was analyzed by LC/MS using a nanoLC (Ultimate 3000, Thermo Scientific) and Orbitrap Fusion Tribrid mass spectrometer (Thermo Scientific). Tandem MS spectra were acquired in a data-dependent manner, where fragmentation was performed by the high-energy collision-induced dissociation method (normalized collision energy = 35). Raw data was converted to an mgf file and applied to a database search using Mascot (Ver. 2.6.2, Matrix Science) and the UniprotKB protein sequence database (42,431 entries). Search conditions were as follows: maximum miss cleavage, 2; peptide mass tolerance, 7 ppm; fragment mass tolerance, 0.02 Da; target false discovery rate, 1%; fixed modification, carbamidomethyl (C); and variable modifications, ammonia-loss (N-terminal C), Gln>pyroGlu (N-terminal Q), and oxidation (M).

## Clinical samples and lectin staining

Based on lectin microarray analysis, WFA was chosen for lectin staining. We investigated patients with HR-positive breast cancer who underwent curative surgery at our hospital from December 2012 to April 2018, inclusive, and received TAM as adjuvant therapy. There were 43 patients who developed distant metastasis during five-year-TAM treatment and 20 of these patients were randomly selected for lectin staining. As a control, 20 patients who received the same treatment but were free from distant metastasis were also analyzed. Clinicopathological features of the 40 patients are shown in S1 Table.

For lectin staining, WFA was biotinylated using a Biotin Labeling Kit-NH2 (Dojindo). Formalin-fixed and paraffin-embedded surgical specimens were cut into 3 μm-thick slices and transferred to an automated staining apparatus, BenchMark GX (Ventana Medical Systems). Tissues were incubated with 10 μg/mL biotinylated WFA. Then, sections were stained with diaminobenzidine (Ventana Medical Systems) and counterstained with hematoxylin (Ventana Medical Systems). WFA staining in cancer cells was semi-quantitatively evaluated. We defined WFA as being positive when more than 10% of cancer cells showed staining for this lectin in the cytoplasm and/or the membrane. Among WFA-positive tumors, the staining was then defined as weak positive or strong positive, according to intensity. If both strongly and weakly-stained cells were observed, the dominant intensity was selected. Representative images are shown in S2 Fig.

## Ethical approval and informed consent

This study was carried out with approval from the Ethics Committee of Juntendo University Hospital (H19-289) and complies with the 1964 Helsinki Declaration and its later amendments or comparable ethical standards. All participants were informed that the research policy was

available on the homepage of the hospital and that they had the opportunity to opt-out of the study at any time later on, which was approved by the Ethics Committee. The Ethics Committee approved of the opt-out method for the use of specimens and clinical data under the condition that all data were anonymized. Only those participants who had not opted-out from the study were included in the data analysis.

## Statistical analysis

Statistical analyses were performed using JMP 14.2 statistical software (SAS Institute, Inc.). Comparisons of mean values between two groups were performed on unpaired data employing the 2-sided $t$ test. A p-value $< 0.050$ was considered to indicate a statistically significant difference.

## Results

### Identification of lectins related to TAM resistance in breast cancer

The membrane protein fraction (hydrophobic) was prepared from cell pellets, and analyzed with the lectin microarray. Among the 45 lectins on the lectin microarray (S3 Fig), the relative intensities of WFA and SNA were higher in T47D-TAM$^R$ (p = 0.002, p = 0.125, respectively) and ZR75-1-TAM$^R$ cells (p < 0.001, p < 0.001, respectively) than in the TAM-sensitive cells (Fig 1). WFA is known to recognize glycan structures containing terminal N-acetylgalactosamine (GalNAc) or terminal galactose (Gal). SNA is known to recognize sialic acid, which is attached to Gal/GalNAc through α2–6 linkage. To verify the results obtained from the lectin microarray, we performed lectin blotting using WFA and SNA lectins. This confirmed the increased binding intensity of WFA in T47D-TAM$^R$ and ZR75-1-TAM$^R$ cells compared with sensitive cells; however, SNA binding intensity was not different between the TAM-sensitive and resistant cells (Fig 2). These results suggest that altered glycosylation in breast cancer cells, which can be detected by WFA, could be related to resistance to TAM therapy.

### Identification of candidate glycoproteins in T47D-TAM$^R$ and ZR75-1-TAM$^R$ that bind WFA

Using lectin microarrays and lectin blotting, we decided to focus on the WFA-binding molecules implicated in TAM resistance of breast cancer cells. Lectin blotting after SDS-PAGE

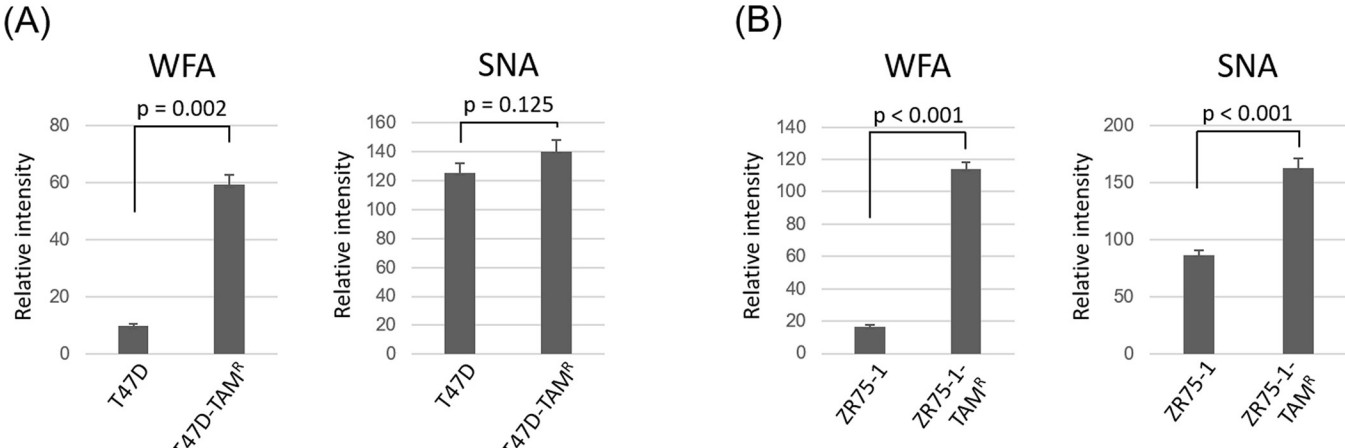

**Fig 1. Lectin microarray analysis of T47D and ZR75-1 cells and their TAM-resistant derivative T47D-TAM$^R$ and ZR75-1-TAM$^R$ cells.** The relative intensities of WFA and SNA in (A) T47D (sensitive) and T47D-TAM$^R$ (TAM-resistant) cells and (B) ZR75-1 (sensitive) and ZR75-1-TAM$^R$ (TAM-resistant) cells, based on normalized average data. Error bars indicate standard deviations. $t$ tests were employed for statistical comparisons.

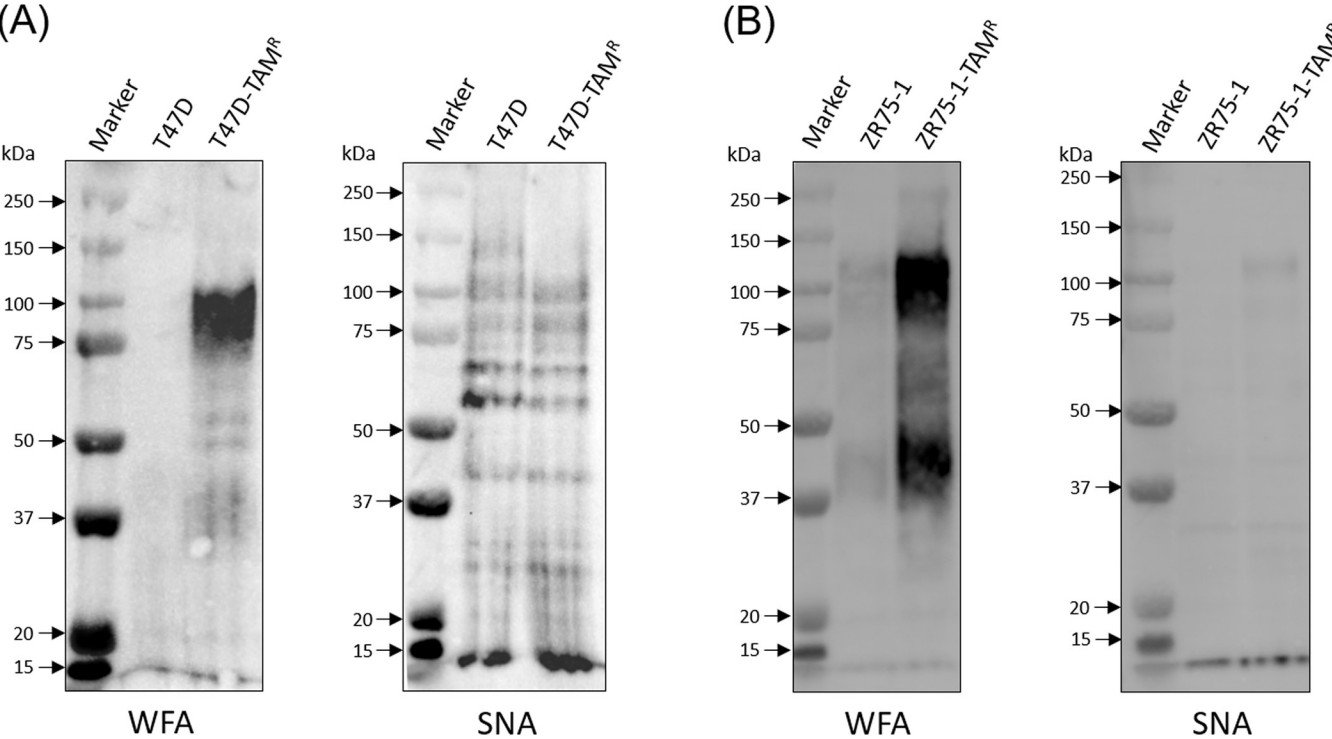

**Fig 2. Lectin blotting analysis of T47D and ZR75-1 cells and their corresponding TAM-resistant cells.** The hydrophobic fractions from sensitive and resistant cells were electrophoretically separated and the binding of WFA and SNA were examined. (A) The results of lectin blotting in T47D and T47D-TAM[R] cells. (B) The results of lectin blotting in ZR75-1 and ZR75-1-TAM[R] cells.

identified WFA binding molecules. From the migration position, the molecular mass of the major component had an approximate molecular mass of 100 kDa (Fig 2). The hydrophobic fraction of T47D-TAM[R] and ZR75-1-TAM[R] cells was subjected to WFA-lectin precipitation and then analyzed by lectin blotting after SDS-PAGE. WFA binding molecules migrated around 100 kDa (Fig 3A). To identify the candidate molecules that bind to WFA, the WFA-bound fraction was separated by SDS-PAGE and the gel was silver stained (Fig 3B). The gel around 100 kDa was excised and subjected to in-gel tryptic digestion. Peptides extracted from the gel were analyzed by LC/MS. Identified proteins are shown in Table 1, which included CD166 and integrin beta-1.

## WFA staining of clinical samples

We conducted WFA staining of surgical specimens, and the relationship between binding intensity and patient outcomes was examined. The majority of primary tumors were positive for WFA both in the relapsed group (n = 20) and the control group who were free from recurrent disease (n = 20), with 19 tumors (95%) and 17 tumors (85%) showing strong/weak positivity, respectively (Fig 4). Interestingly, strong WFA staining was more frequently observed in the relapsed group, who developed distant metastasis during TAM treatment, compared with those in the control group (Fig 4, p = 0.011). We further conducted WFA staining in metastatic lesions to compare with primary tumors. Among the 20 metastatic patients, a biopsy of the metastatic lesion was performed in five patients and three of these cases were available for WFA staining. In these three cases, there was no difference in WFA expression between primary and metastatic lesions as WFA was already strongly positive in primary tumors and was maintained in the metastatic lesion (S4 Fig).

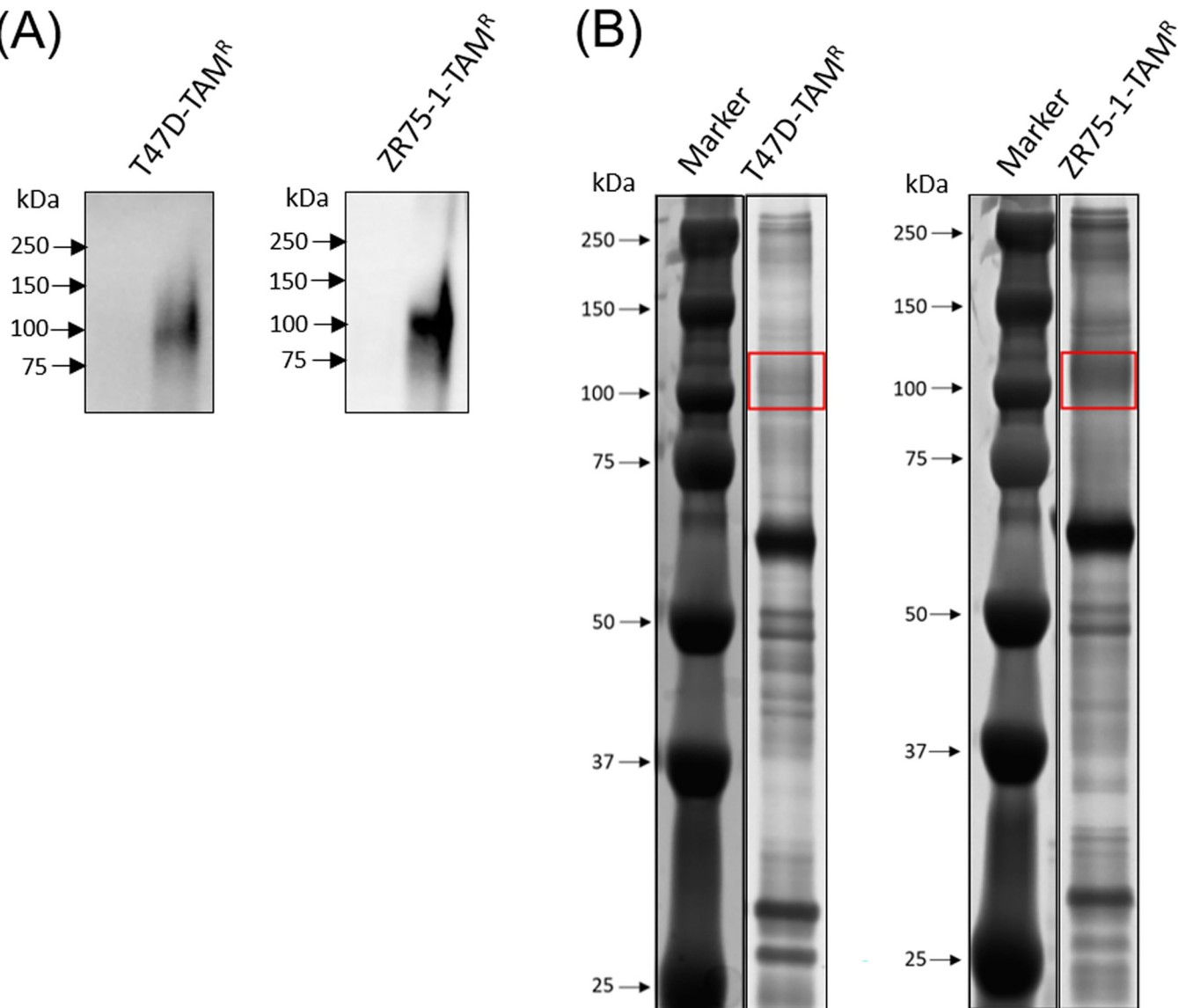

**Fig 3. Precipitation of WFA binding glycoproteins in T47D-TAM[R] and ZR75-1-TAM[R] cells.** (A) Blotting of WFA binding fractions with WFA. (B) Silver staining of WFA binding fractions. Red rectangles indicate bands around 100 kDa, which were cut out and used for mass spectrometry.

## Discussion

Despite early detection techniques, therapeutic regimens and an understanding of the molecular basis of breast cancer biology, more than 30% of patients receiving adjuvant ET develop metastatic disease, during or after treatment [19]. It is critical to identify this group of patients and to design new therapeutic strategies suited to this subset.

Although altered cellular glycosylation is known to be associated with cancer progression and metastasis [11], reports on altered glycosylation in cancer cells resistant to anticancer drugs are limited. One important aspect is that glycosylation is known to modulate the function of molecules involved in apoptosis [20]. We recently reported that Mucin 21 confers resistance to apoptosis in an O-glycosylation-dependent manner [21]. Another aspect is that the function of drug transporters is glycosylation-dependent [22–24], with studies showing that

**Table 1. Identified candidate proteins in the WFA binding fraction (100 kDa gel) in T47D-TAM^R and ZR75-1-TAM^R cells.**

| T47D-TAM^R | | | | | | |
| --- | --- | --- | --- | --- | --- | --- |
| Number of identified peptides/ accession (WFA binding fraction) | Number of identified peptides/ accession (negative control) | Protein accession | Gene Name | Protein description | Protein mass (Da) | Length (amino acids) |
| 37 | n.i.* | Q14697 | GANAB | Neutral alpha-glucosidase AB | 107.263 | 944 |
| 37 | n.i.* | | GANAB | Isoform 2 of Neutral alpha-glucosidase AB | 109.825 | 966 |
| 19 | n.i.* | P55060 | CSE1L | Exportin-2 | 111.145 | 971 |
| 18 | n.i.* | Q13740 | ALCAM | CD166 antigen | 65.745 | 583 |
| 15 | n.i.* | P07900 | HSP90AA1 | Heat shock protein HSP 90-alpha | 85.006 | 732 |
| 13 | n.i.* | P13639 | EEF2 | Elongation factor 2 | 96.246 | 858 |
| 13 | n.i.* | P14625 | HSP90B1 | Endoplasmin | 92.696 | 803 |
| 12 | n.i.* | P02538 | KRT6A | Keratin, type II cytoskeletal 6A | 60.293 | 564 |
| 12 | n.i.* | P05556 | ITGB1 | Integrin beta-1 | 91.664 | 798 |
| 8 | n.i.* | P19367 | HK1 | Hexokinase-1 | 103.561 | 917 |
| 8 | n.i.* | Q14525 | KRT33B | Keratin, type I cuticular Ha3-II | 47.325 | 404 |
| 7 | n.i.* | P35221 | CTNNA1 | Catenin alpha-1 | 100.693 | 906 |
| 7 | n.i.* | Q15323 | KRT31 | Keratin, type I cuticular Ha1 | 48.633 | 416 |
| 6 | n.i.* | P13646 | KRT13 | Keratin, type I cytoskeletal 13 | 49.900 | 458 |
| 5 | n.i.* | P04792 | HSPB1 | Heat shock protein beta-1 | 22.826 | 205 |
| 5 | n.i.* | P32004 | L1CAM | Neural cell adhesion molecule L1 | 140.885 | 1257 |
| 5 | n.i.* | Q14533 | KRT81 | Keratin, type II cuticular Hb1 | 56.832 | 505 |
| ZR75-1-TAM^R | | | | | | |
| 30 | n.i.* | Q13740 | ALCAM | CD166 antigen | 65.745 | 583 |
| 13 | n.i.* | P05556 | ITGB1 | Integrin beta-1 | 91.664 | 798 |
| 4 | n.i.* | O43493 | TGOLN2 | Trans-Golgi network integral membrane protein 2 | 50.988 | 479 |
| 3 | n.i.* | P19440 | GGT1 | Glutathione hydrolase 1 proenzyme | 61.714 | 569 |
| 3 | n.i.* | P36268 | GGT2 | Inactive glutathione hydrolase 2 | 62.074 | 569 |
| 3 | n.i.* | | GGT2 | Inactive glutathione hydrolase 2 | 61.026 | 559 |
| 3 | n.i.* | | GGT2 | Isoform 3 of Inactive glutathione hydrolase 2 | 62.538 | 574 |
| 3 | n.i.* | Q14697 | GANAB | Neutral alpha-glucosidase AB | 107.263 | 944 |
| 3 | n.i.* | | GANAB | Isoform 2 of Neutral alpha-glucosidase AB | 109.825 | 966 |
| 3 | n.i.* | Q9Y639 | NPTN | Neuroplastin | 44.702 | 398 |
| 3 | n.i.* | | NPTN | Isoform 1 of Neuroplastin | 31.500 | 282 |
| 3 | n.i.* | | NPTN | Isoform 3 of Neuroplastin | 31.044 | 278 |
| 3 | n.i.* | | NPTN | Isoform 4 of Neuroplastin | 38.110 | 337 |
| 3 | n.i.* | | NPTN | Isoform 5 of Neuroplastin | 44.245 | 394 |
| 2 | n.i.* | A6NGU5 | GGT3P | Putative glutathione hydrolase 3 proenzyme | 61.919 | 568 |
| 2 | n.i.* | P05090 | APOD | Apolipoprotein D | 21.547 | 189 |
| 2 | n.i.* | P14625 | HSP90B1 | Endoplasmin | 92.696 | 803 |

WFA, *Wisteria floribunda* agglutinin; TAM, tamoxifen; n.i., not identified

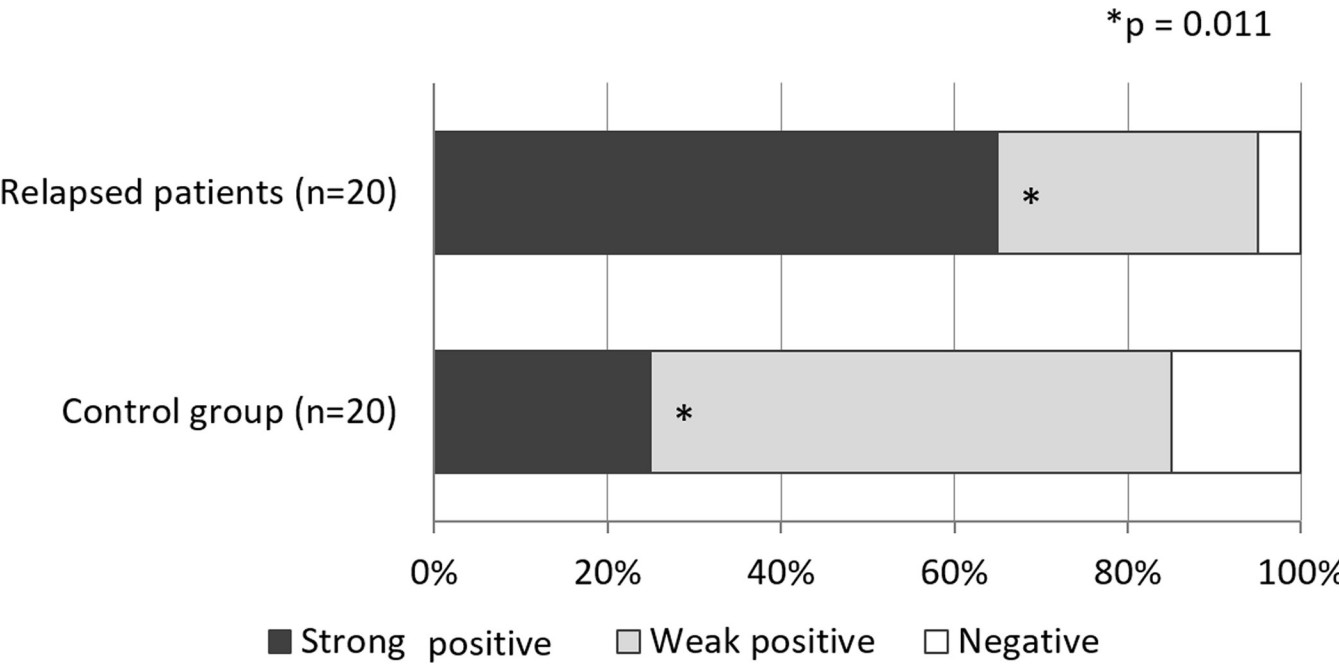

**Fig 4. Histochemical WFA staining of sections from primary breast cancer specimens.** Primary tumors from patients who developed distant metastasis during adjuvant TAM treatment (n = 20) and those who did not develop recurrent disease (n = 20) were analyzed. Strong positive staining (dark grey) was significantly more frequent in relapsed patients (13 patients, 65%) than in the control group (4 patients, 20%; p = 0.011).

different glycans influence the sensitivity to different drugs through a variety of mechanisms. In breast cancer, although the importance of glycosylation as a biomarker has been extensively investigated [25], there have been no previous studies regarding glycan-based analyses in TAM-resistant luminal type breast cancer cells. In the present study, we performed lectin microarray analysis of membrane fractions of newly developed TAM-resistant breast cancer cells. We found increased WFA-binding in TAM-resistant breast cancer cells, compared with TAM-sensitive cells.

Studies investigating the relationships between serum biomarkers and various diseases have reported the usefulness of WFA binding [26–28]. For instance, Fujiyoshi *et al.* reported elevated serum WFA-positive Mac-2-binding protein levels were a significant risk factor for tumor recurrence in hepatocellular carcinoma [27]. Most of these previous studies have focused on the changes in WFA binding in serum samples; few studies have focused on WFA binding to cell surface molecules [29]. We previously reported that surgical specimens of relapsed triple-negative breast cancer patients showed higher levels of WFA binding than that of non-relapsed groups, using lectin microarrays [29]. In the present study, we identified WFA as a lectin having differential binding between TAM-resistant and sensitive breast cancer cells. In addition, we found strong WFA staining in the relapsed patients treated with TAM.

Recently, WFA was shown to strongly interact with the LacdiNAc structure produced by human cervical carcinoma cells, and 1,3-*N*-acetylgalactosaminyltransferase 2 (B3GALNT2) is responsible for its biosynthesis [30]. Using publicly available Kaplan–Meier plotter mRNA microarray databases [31], we examined the relationship between *B3GALNT2* expression and the recurrence-free survival of patients who were given adjuvant TAM treatment. Patients with tumors containing high levels of B3GALNT2 mRNA exhibited significantly shorter recurrence-free periods than those with low levels of mRNA for this enzyme (S5 Fig). If the mRNA levels correspond to the formation of WFA-reactive glycans, then these indirect

database results can be considered consistent with the lectin microarray data and lectin staining data we obtained.

Further investigations are necessary to determine the mechanism of how WFA-binding glycans contribute to TAM-resistance. Che *et al.* reported that WFA-binding glycoproteins promoted cancer stemness via epidermal growth factor receptor signaling in colorectal cancer [32]. A similar mechanism may be responsible for therapeutic resistance in hormone-positive breast cancer. However, such mechanisms should be explored after the identification of the glycoproteins carrying WFA-binding glycans. Asif *et al.* reported that activated integrin beta-1 is increased in metastatic breast cancer cells [33], and CD166 is also upregulated in TAM-resistant breast cancer cells [34]. Interestingly, in our present study, integrin beta-1 and CD166 were identified as candidates for WFA binding membrane glycoproteins. However, we could not verify whether the amount of these glycoproteins themselves was greater in resistant cells compared with sensitive cells, or whether structural differences in the glycans of these glycoproteins on the TAM-resistant cells made them reactive with WFA.

In clinical samples, strong WFA staining was more frequently observed in the relapsed patients, compared with non-relapsed patients, indicating that this lectin might have the potential to be a predictive marker for *de novo* resistance to TAM treatment, not just acquired resistance. When comparing WFA staining intensity in metastatic lesions that developed during TAM treatment, there was no difference in the expression level compared with their paired primary lesion, although we assessed only three patients. Further studies with larger samples are needed to confirm this finding. Moreover, to avoid inter- and intra-observational differences and obtain more reproducible results, another more detailed scoring system should be considered in future studies, as suggested by some recent reports [35, 36].

In conclusion, we found that WFA, which is reported to recognize the LacdiNAc structure containing terminal GalNAc residues, showed stronger binding to TAM-resistant breast cancer cells. Moreover, strong WFA staining was observed in patients who developed distant metastasis during TAM treatment. Our results suggest that further investigation is warranted into glycoproteins with WFA-binding glycans on TAM resistant HR-positive breast cancer cells.

## Supporting information

**S1 Fig. Cell proliferation assays in T47D, ZR75-1 and TAM-resistant cells.** Cell viability of T47D, T47D-TAM$^R$, ZR75-1 and ZR75-1-TAM$^R$ cells after 72 h of TAM treatment. The LD50 of T47D and T47D-TAM$^R$ cells was 4.0 μM and 6.3 μM, respectively, and that of ZR75-1 and ZR75-1-TAM$^R$ cells was 9.5 μM and 10.4 μM, respectively. Error bars indicate standard deviations.
(PDF)

**S2 Fig. Representative images of HR-positive breast cancer tissues stained with WFA.** WFA staining was assessed as negative, weak positive or strongly positive. Strong staining was observed mainly in the cell membrane and cytoplasm of HR-positive breast cancer tissues.
(PDF)

**S3 Fig. Lectin microarray analysis.** Lectin microarray analysis of (A) T47D and T47D-TAM$^R$ cells and (B) ZR75-1 and ZR75-1-TAM$^R$ cells, showing the relative intensities of 45 lectins in the TAM-sensitive and resistant cells, based on normalized average data. Error bars indicate standard deviations.
(PDF)

**S4 Fig. Comparison of WFA staining in primary and metastatic lesions.** WFA staining of primary and metastatic lesions in three patients are shown. Histological type of patient 2 was

mucinous carcinoma. Metastatic lesions of patient 1, 2 and 3 are duodenum, lung, and liver, respectively.
(PDF)

**S5 Fig. Kaplan-Meier curves of recurrence-free-survival according to *B3GALNT2* mRNA expression (n = 178).** The curve shown in red represents recurrence-free-survival of patients with *B3GALNT2* mRNA-high tumors, while the black curve shows those with low *B3GALNT2* tumors. The data was obtained from publicly available Kaplan–Meier plotter mRNA microarray databases [29].
(PPTX)

**S1 Raw images. Raw images of Fig 2 and Fig 3.**
(PDF)

**S1 Table. Clinicopathological features of patients.**
(XLSX)

## Acknowledgments

The authors sincerely appreciate Clear Science Pty Ltd. for language editing. We also thank the members of the Laboratory of Morphology and Image Analysis and Research Support Center, Juntendo University Graduate School of Medicine, for technical assistance.

## Author Contributions

**Conceptualization:** Yoshiya Horimoto, Yumiko Ishizuka, Tatsuro Irimura.

**Data curation:** Yoshiya Horimoto, Yumiko Ishizuka, Mitsue Saito.

**Investigation:** May Thinzar Hlaing, Yoshiya Horimoto, Kaori Denda-Nagai, Haruhiko Fujihira, Miki Noji, Hiroyuki Kaji, Azusa Tomioka, Yumiko Ishizuka, Harumi Saeki, Atsushi Arakawa.

**Methodology:** Yoshiya Horimoto, Kaori Denda-Nagai, Haruhiko Fujihira, Hiroyuki Kaji, Tatsuro Irimura.

**Project administration:** Yoshiya Horimoto, Tatsuro Irimura.

**Resources:** Hiroyuki Kaji, Atsushi Arakawa, Mitsue Saito, Tatsuro Irimura.

**Supervision:** Mitsue Saito, Tatsuro Irimura.

**Validation:** May Thinzar Hlaing, Kaori Denda-Nagai, Miki Noji, Hiroyuki Kaji, Azusa Tomioka.

**Visualization:** May Thinzar Hlaing, Yoshiya Horimoto.

**Writing – original draft:** May Thinzar Hlaing, Yoshiya Horimoto, Haruhiko Fujihira.

**Writing – review & editing:** Hiroyuki Kaji, Tatsuro Irimura.

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
