## [Decision Letter · Decision Letter 0]

13 Jun 2022

PONE-D-22-13506Glycans recognized by Wisteria floribunda agglutinin as a potential marker for resistance against endocrine treatment in breast cancerPLOS ONE

Dear Dr. Yoshiya Horimoto,

Thank you for submitting your manuscript to PLOS ONE. After careful consideration, we feel that it has merit but does not fully meet PLOS ONE’s publication criteria as it currently stands. Therefore, we invite you to submit a revised version of the manuscript that addresses the points raised during the review process.

Please see below comments and observations made by the reviewers and myself.

We look forward to receiving your revised manuscript.

Kind regards,

Patricia Talamas-Rohana, Ph.D.

Academic Editor

PLOS ONE

Journal Requirements:

3. PLOS ONE now requires that submissions reporting blots or gels include original, uncropped blot/gel image data as a supplement or in a public repository. This is in addition to complying with our image preparation guidelines described at https://journals.plos.org/plosone/s/figures#loc-blot-and-gel-reporting-requirements. These requirements apply both to the main figures and to cropped blot/gel images included in Supporting Information. If the manuscript is positively reviewed, we will ask the authors to provide any missing raw image data for blot/gel results when they submit their first revision. As part of your review, please ensure that figures reporting blot or gel images comply with the journal’s image preparation guidelines and that the original data are provided following the journal’s request.  If you have any questions or concerns about blot/gel figures or data for this submission, please email us at plosone@plos.org before issuing a decision letter.

Additional Editor Comments:

I have recommended major revision for your manuscript, based on the comments and observations made by three reviewers.

Although your manuscript presents primary research that contributes to scientific knowledge, it requires additional work because publication criteria 3 and 4 were not completely covered.

Criteria No. 3. Experiments, statistics, and other analyses are performed to a high technical standard and are described in sufficient detail.

The three reviewers are asking for further methodological descriptions. Moreover, I consider that the inclusion of the MS analysis of sensitive cells should be included as control.

In addition, biofinformatic analysis can be performed in order to determine whether the identified proteins contain the N-acetylgalactosamine beta 1 (GalNAc beta 1-3 Gal) modification that is recognized by the WFA lectin.

Criteria No. 4. Conclusions are presented in an appropriate fashion and are supported by the data.

The manuscript does not contain any experimental results that confirm the role of the identified glycoproteins in the TMX resistance mechanism. For this reason, reviewer No. 1 rejected the manuslcript and reviewer No. 3 is proposing a change in the title of the manuscript. Therefore, authors have to decide if they go deeper in looking for the participation of the glycoproteins identified in the resistance mechanisms or discard this asseveration from the title.

Reviewers' comments:

Reviewer's Responses to Questions

**Comments to the Author**

1. Is the manuscript technically sound, and do the data support the conclusions?

Reviewer #1: No

Reviewer #2: Yes

Reviewer #3: Partly

2. Has the statistical analysis been performed appropriately and rigorously? 

Reviewer #1: No

Reviewer #2: I Don't Know

Reviewer #3: No

3. Have the authors made all data underlying the findings in their manuscript fully available?

Reviewer #1: No

Reviewer #2: No

Reviewer #3: Yes

4. Is the manuscript presented in an intelligible fashion and written in standard English?

Reviewer #1: No

Reviewer #2: Yes

Reviewer #3: Yes

5. Review Comments to the Author

Reviewer #1: The topic on glycans for drug resistant in breast cancer treamtent is interesting. However, the data

current can not support the finding and conculsions.

1. Glycomics has been studied in other reports, where the modification sites and glycan structures

can be clearly identified. The authors carried out mass spectrometry analysis but did not give out any

meaningful proteomic data (probably only identify some protein sequences)

2.Lectin array can only give the subtype of glycoprotein expression and it should be integrated with

protein expression (WB) to see the glycosylation change.

3. Clinical sample analysis is not very related cell-based study, which need more data to support (e.g. quantitative glycoproteomics on the clinical samples).

Reviewer #2: Line 94: resistant cell lines are described as if they had been developed in the laboratory itself and this is not the case, they were acquired from a commercial company.

Line 100: the term "parental cells" does not apply, although it is the same cell line with acquired resistance they do not come from the same cell passage

Line 256: the proteins identified in the WFA-binding fraction in TAM-cells are mentioned, nevertheless the same analysis should be carried out in the cells not TAM to be able to conclude with greater support the results

Line 264: It is not clear whether the control group refers to patients treated with tamoxifen who did not develop metastases or rather who did not relapse.

Line 307: is incorrectly worded, lectin binding to glycoproteins does not contribute to resistance, but rather could function as a potential biomarker or prognostic factor.

Reviewer #3: In the methods section, the identification of lectin-bound proteins should be mentioned in a single section.

In the methods section, the quantification of the biotinylated lectin reaction in tissue from cancer patients is deficient. It is suggested to explain in detail. Consider published methods to assess different levels of positivity.

1) Diagn Pathol. 2014 Nov 29;9:221.doi: 10.1186/s13000-014-0221-9.Different approaches for interpretation and reporting of immunohistochemistry analysis results in the bone tissue - a review. Nickolay Fedchenko 1 2, Janin Reifenrath 3. DOI: 10.1186/s13000-014-0221-9

2) Pathol Res Pract. 2015 Dec;211(12):973-81.doi: 10.1016/j.prp.2015.10.002. Epub 2015 Nov 6. Integrins and haptoglobin: Molecules overexpressed in ovarian cancer. Julio César Villegas-Pineda 1, Olga Lilia Garibay-Cerdenares 2, Verónica Ivonne Hernández-Ramírez 3, Dolores Gallardo-Rincón 4, David Cantú de León 5, María Delia Pérez-Montiel-Gómez 6, Patricia Talamás-Rohana 7

3) Cancer Cell Int. 2022; 22: 6. doi: 10.1186/s12935-021-02425-6 PHD finger protein 20-like protein 1 (PHF20L1) in ovarian cancer: from its overexpression in tissue to its upregulation by the ascites microenvironment. Dulce Rosario Alberto-Aguilar,1 Verónica Ivonne Hernández-Ramírez,1 Juan Carlos Osorio-Trujillo,1 Dolores Gallardo-Rincón,2 Alfredo Toledo-Leyva,2 and Patricia Talamás-Rohana1

The title does not match the results. Although the work demonstrates that the WFA lectin is capable of binding to proteins expressed in cancer cells resistant to TAM, it is recommended to carry out a statistical correlation analysis between positivity with WFA and the presence of resistance to TAM, and/or the clinical- pathological characteristics of the patients. One option is to change the job title, the suggestion would be:

TAM-resistant breast cancer cells react positively to the lectin WFA.

6. PLOS authors have the option to publish the peer review history of their article (what does this mean?). If published, this will include your full peer review and any attached files.

Reviewer #1: No

Reviewer #2: No

Reviewer #3: **Yes: **Veronica Ivonne Hernández Ramírez

---

## [Author Response · Author response to Decision Letter 0]

5 Aug 2022

July 07, 2022

Dear Editor,

We very much appreciate the Reviewers’ and your careful reading and evaluation of our manuscript, as well as your invitation to resubmit our work after appropriate revision. Based on all your comments, we have revised our manuscript and are re-submitting it herewith. Our point-by-point responses to all your comments are provided below, highlighted in Arial font. All revisions in the text are in red font.

Response to Editors and Reviewers:

Reviewer #1: The topic on glycans for drug resistant in breast cancer treatment is interesting. However, the data current cannot support the finding and conclusions.

1. Glycomics has been studied in other reports, where the modification sites and glycan structures

can be clearly identified. The authors carried out mass spectrometry analysis but did not give out any

meaningful proteomic data (probably only identify some protein sequences).

>As the Reviewer mentioned, for example, a study that we cited as reference 16 (J Proteomics 2021) revealed data on changes in glycan structure observed in paclitaxel-resistant cells. However, to our knowledge, hormone therapy resistance has not been investigated in detail. Nevertheless, the purpose of our study was not to reveal meaningful proteomic data, but to identify differences in glycan expression in TAM-resistant cells. We apologize for any misleading expressions in the text. We have corrected some descriptions in the Abstract and Discussion section to avoid confusion among readers. We hope these comments and revisions will satisfy the Editors and Reviewers.

2. Lectin array can only give the subtype of glycoprotein expression and it should be integrated with

protein expression (WB) to see the glycosylation change.

>Our data could mean either a change in glycosylation or a change in expression of the glycoproteins themselves. To further investigate these points, expression and binding by immunoprecipitation with the lectins and candidate proteins should be tested. We tried several commercially available antibodies for immunoprecipitations and western blot but unfortunately they did not work well in these experiments. Therefore, the Reviewer's point is yet to be answered, but we still believe that our current data will be of interest to your readers.

3. Clinical sample analysis is not very related cell-based study, which need more data to support (e.g. quantitative glycoproteomics on the clinical samples).

>As mentioned above (question 1), we could not manage to identify specific glycoprotein(s). Therefore, further clinical sample analysis, which was suggested by the Reviewers, could not be performed. Glycoproteomic analysis of clinical samples is a subject of our future work and will hopefully be published as a separate paper.

Reviewer #2:

Line 94: resistant cell lines are described as if they had been developed in the laboratory itself and this is not the case, they were acquired from a commercial company.

>We apologize for our confusing explanation. The resistant cells were established in our laboratory. We have corrected the description in the Methods section.

Line 100: the term "parental cells" does not apply, although it is the same cell line with acquired resistance, they do not come from the same cell passage

>We have now revised our manuscript throughout. “Parental/parent” has been replaced with “sensitive” although there may be some confusion by the reviewer as stated above.

Line 256: the proteins identified in the WFA-binding fraction in TAM-cells are mentioned, nevertheless the same analysis should be carried out in the cells not TAM to be able to conclude with greater support the results

>We appreciate the reviewer’s suggestion. Binding to WFA was elevated only in TAM-resistant cells, and it was not possible to extract WFA-binding glycoproteins by affinity-isolation with WFA from TAM-sensitive cells. We realized that there were some common proteins detected in the two TAM-resistant cell lines. Therefore, it was not possible to carry out “the same analysis” with TAM-sensitive cells.

Line 264: It is not clear whether the control group refers to patients treated with tamoxifen who did not develop metastases or rather who did not relapse.

>We meant those who did not develop metastases. We have now revised the sentence to “Primary tumors from patients who developed distant metastasis during adjuvant TAM treatment (n = 20) and those who did not develop recurrent disease (n = 20) were analyzed”.

Line 307: is incorrectly worded, lectin binding to glycoproteins does not contribute to resistance, but rather could function as a potential biomarker or prognostic factor.

>We appreciate the Reviewer pointing this out. We have now revised the statement to “When comparing WFA staining intensity in metastatic lesions that developed during TAM treatment, there was no difference in the expression level compared with their paired primary lesion…”.

Reviewer #3: In the methods section, the identification of lectin-bound proteins should be mentioned in a single section.

>We appreciate the Reviewer’s suggestion. We have now revised the Methods section under a new title “Preparation of WFA-binding proteins and identification by liquid chromatography/mass spectrometry (LC/MS) followed by database search using Mascot”.

In the methods section, the quantification of the biotinylated lectin reaction in tissue from cancer patients is deficient. It is suggested to explain in detail. Consider published methods to assess different levels of positivity.

1) Diagn Pathol. 2014 Nov 29;9:221.doi: 10.1186/s13000-014-0221-9.Different approaches for interpretation and reporting of immunohistochemistry analysis results in the bone tissue - a review. Nickolay Fedchenko 1 2, Janin Reifenrath 3. DOI: 10.1186/s13000-014-0221-9

2) Pathol Res Pract. 2015 Dec;211(12):973-81.doi: 10.1016/j.prp.2015.10.002. Epub 2015 Nov 6. Integrins and haptoglobin: Molecules overexpressed in ovarian cancer. Julio César Villegas-Pineda 1, Olga Lilia Garibay-Cerdenares 2, Verónica Ivonne Hernández-Ramírez 3, Dolores Gallardo-Rincón 4, David Cantú de León 5, María Delia Pérez-Montiel-Gómez 6, Patricia Talamás-Rohana 7

3) Cancer Cell Int. 2022; 22: 6. doi: 10.1186/s12935-021-02425-6 PHD finger protein 20-like protein 1 (PHF20L1) in ovarian cancer: from its overexpression in tissue to its upregulation by the ascites microenvironment. Dulce Rosario Alberto-Aguilar,1 Verónica Ivonne Hernández-Ramírez,1 Juan Carlos Osorio-Trujillo,1 Dolores Gallardo-Rincón,2 Alfredo Toledo-Leyva,2 and Patricia Talamás-Rohana1

>We appreciate the reviewer’s suggestion. First, we simply semi-quantitatively assessed the WFA expression in surgical specimens without relying on any scoring system. We have now added the necessary description in the Methods section and legends for Figure 4. 

We thank the reviewer for giving us useful information from some previous studies. Unfortunately we could not manage to conduct such a detailed assessment as that performed by Dulce Rosario Alberto-Aguilar et al (Cancer Cell Int. 2022), but now we are well aware of the need to perform further studies to obtain more quantitative results. We have now added this issue as a limitation of the current study in the Discussion section (lines 315-317). We hope these comments and revisions will satisfy the Editors and Reviewers.

The title does not match the results. Although the work demonstrates that the WFA lectin is capable of binding to proteins expressed in cancer cells resistant to TAM, it is recommended to carry out a statistical correlation analysis between positivity with WFA and the presence of resistance to TAM, and/or the clinical- pathological characteristics of the patients. One option is to change the job title, the suggestion would be: TAM-resistant breast cancer cells react positively to the lectin WFA.

>We appreciate the reviewer’s suggestion. We have now changed the title as suggested. Because the term WFA (Wisteria floribunda agglutinin) includes “lectin”, we decided to use the revised title “TAM-resistant breast cancer cells exhibit reactivity with Wisteria floribunda agglutinin”.

Additional Editor Comments:

I have recommended major revision for your manuscript, based on the comments and observations made by three reviewers. Although your manuscript presents primary research that contributes to scientific knowledge, it requires additional work because publication criteria 3 and 4 were not completely covered.

Criteria No. 3. Experiments, statistics, and other analyses are performed to a high technical standard and are described in sufficient detail.

The three reviewers are asking for further methodological descriptions. Moreover, I consider that the inclusion of the MS analysis of sensitive cells should be included as control.

In addition, bioinformatic analysis can be performed in order to determine whether the identified proteins contain the N-acetylgalactosamine beta 1 (GalNAc beta 1-3 Gal) modification that is recognized by the WFA lectin.

>Methodological descriptions have been revised accordingly.

We judged it was not possible to conduct MS analysis of the sensitive cells as a control, due to the very low binding level of WFA in the sensitive cells, as shown in Figure 2.

We appreciate your suggestion of further bioinformatic analysis. However, we were not able to identify the specific glycoproteins that bind to WFA in this study. Therefore, we could not perform the analysis you suggested. We investigated the relationship between the expression of B3GALNT2, a GalNAcbeta1-3GlcNAc glycosyltransferase which is responsible for the biosynthesis of the LacdiNAc structure having high affinity with WFA (reference 28), and the prognosis of patients who were treated with TAM, employing a data base available in the public domain (Kaplan-Meier plotter). The results showed that patients with B3GALNT2-high tumors had significantly worse outcomes. The results of this analysis have been added as Supplementary Figure 5 and have been mentioned in the revised Discussion. Although these indirect observations should be interpreted with caution, we believe that the specific glycan structures recognized by WFA merit further investigation in relation to resistance to TAM. We hope these comments and revisions will satisfy the Editors.

Criteria No. 4. Conclusions are presented in an appropriate fashion and are supported by the data.

The manuscript does not contain any experimental results that confirm the role of the identified glycoproteins in the TMX resistance mechanism. For this reason, reviewer No. 1 rejected the manuscript and reviewer No. 3 is proposing a change in the title of the manuscript. Therefore, authors have to decide if they go deeper in looking for the participation of the glycoproteins identified in the resistance mechanisms or discard this asseveration from the title.

>We appreciate the Editor’s suggestion. Because we were unable to determine the specific glycoproteins, we have changed the title as suggested.

We again thank you for your consideration. We look forward to hearing from you at your earliest convenience.

Yours sincerely,

Yoshiya Horimoto

---

## [Editor Report · Decision Letter 1]

10 Aug 2022

Tamoxifen-resistant breast cancer cells exhibit reactivity with Wisteria floribunda agglutinin.

PONE-D-22-13506R1

Dear Dr. Horimoto,

We’re pleased to inform you that your manuscript has been judged scientifically suitable for publication and will be formally accepted for publication once it meets all outstanding technical requirements.

Kind regards,

Patricia Talamas-Rohana, Ph.D.

Academic Editor

PLOS ONE
---

## [Editor Report · Acceptance letter]

16 Aug 2022

PONE-D-22-13506R1 

Tamoxifen-resistant breast cancer cells exhibit reactivity with *Wisteria floribunda* agglutinin. 

Dear Dr. Horimoto:

I'm pleased to inform you that your manuscript has been deemed suitable for publication in PLOS ONE. Congratulations! Your manuscript is now with our production department. 

Kind regards, 

on behalf of

Dr. Patricia Talamas-Rohana 

Academic Editor

PLOS ONE